# *Clostridioides difficile* Flagella

**DOI:** 10.3390/ijms25042202

**Published:** 2024-02-12

**Authors:** Jean-Christophe Marvaud, Sylvie Bouttier, Johanna Saunier, Imad Kansau

**Affiliations:** 1Institut MICALIS, INRAE, AgroParisTech, Equipe Bactéries Pathogènes et Santé, Faculté de Pharmacie, Université Paris-Saclay, 91400 Orsay, Franceimad.kansau@universite-paris-saclay.fr (I.K.); 2Matériaux et Santé, Faculté de pharmacie, Université Paris Saclay, 91400 Orsay, France

**Keywords:** *Clostridioides difficile*, flagellum, motility

## Abstract

*Clostridioides difficile* is an important pathogen for humans with a lead in nosocomial infection, but it is also more and more common in communities. Our knowledge of the pathology has historically been focused on the toxins produced by the bacteria that remain its major virulence factors. But the dysbiosis of the intestinal microbiota creating the conditions for the colonization appears to be fundamental for our understanding of the disease. Colonization implies several steps for the bacteria that do or do not use their capacity of motility with the synthesis of flagella. In this review, we focus on the current understanding of different topics on the *C. difficile* flagellum, ranging from its genetic organization to the vaccinal interest in it.

## 1. Introduction

*Clostridioides difficile*, or *C. difficile* (formerly *Clostridium difficile*), is a Gram-positive, anaerobic, multiple-antibiotic-resistant, and spore-forming bacterium found in the intestinal tracts of both humans and animals. *C. difficile* is recognized as the principal pathogen of healthcare-associated diarrhea, and the clinical spectrum of *C. difficile* infection (CDI) ranges from mild diarrhea to severe and life-threatening fulminant colitis resulting from complications such as toxic megacolon and sepsis [1,2].

After a dramatic increase in terms of prevalence and mortality due to the emergence of a hypervirulent strain in the 2000s, the incidence has decreased in the last decade due to preventive measures. However, community-associated CDI has been increasing, accounting for approximately half of all CDI cases in the USA [3].

Under an easily transmittable spore form, *C. difficile* transmission occurs via the fecal–oral route, and the germination of the spores into vegetative cells eventually takes place in the duodenum. Then, the bacteria reach the colon, where they produce one or a combination of toxins. Indeed, depending on the strains, *C. difficile* can produce large toxins, such as TcdA and TcdB, as well as a binary toxin, CDT. TcdA and TcdB induce the disruption of tight junctions and lead to epithelial cell death by the inactivation of host Ras homologous (Rho)-family guanosine triphosphatases (GTPases) [4,5]. Another aspect of the pathogenicity is severe inflammation resulting from the activation of signaling pathways after the recognition of the toxins and flagella by the innate immune system [6,7]. The flagellum is therefore an important factor of pathogenicity during *C. difficile* infection and could exemplify a Gram-positive flagellar organization of an important human pathogen. To confer a fitness advantage to a bacterium, flagellum synthesis should be subject to precise regulation. *C. difficile* is an interesting model of bacteria with many different mechanisms of flagellar regulation that have been described recently. Also, the *C. difficile* flagellum is subject to many studies for its potential biological functions, such as adhesion or biofilm formation during the pathogenicity steps. This could lead to new therapeutic strategies.

In this review, we present the latest data on the genetic organization of the flagellum, its different mechanisms of regulation of synthesis and glycosylation, and its putative role in the pathogenicity of the bacterium, and we conclude with the pertinence of developing a flagella-based vaccine or vaccine adjuvant.

The expression of the flagellar genes in bacteria is temporal and regulated in a hierarchical manner. Thus, in *C. difficile*, the flagellum genetic organization is made of three distinct operons that are temporarily expressed [8]. The early-stage operon, also named F3, is orthologous to the class II flagellar genes found in *Bacillus subtilis*. It is devoted to the assembly of the basal body. It also contains a transcriptional regulator gene encoding a sigma factor similar to SIgD that controls flagellar synthesis and motility and vegetative autolysins in *B. subtilis*. Upstream of this operon is an operon named F2 that encodes flagellar glycan biosynthetic genes, as well as the operon named the late operon, or F1, which notably harbors the flagellin gene *fliC* (Table 1 and Figure 1). The details of the different mechanisms of the regulation of the expression of these operons are discussed below in a specific section.

While the flagellar locus organization is well conserved in *C. difficile* strains, some particularities have been found in few strains. After the sequencing of the genome of the M120 strain (ribotype RT078), the complete loss of the early flagellar region with the retention of the late region was found [9]. When Wu et al. also sequenced the complete whole genomes of three *C. difficile* isolates, they found that a series of genes from *fliP* to *fliM* in the early operon were absent again in the genome of one strain of RT078, resulting in immobility [10]. Also, Androga et al. studied the virulence of the strains of ten different *C. difficile* ribotypes producing only CDT. Strains of two ribotypes had deletions in the F2 and early regions of their flagellum operons, while the F2 region was absent from the strains of seven ribotypes. The flagellin *fliC* and flagellar cap *fliD* genes were conserved in all strains [11]. Until now, the flagellin gene has always been found to be present on the genome of the bacteria and it could be concluded that *C. difficile* is mostly a motile bacterium, but some strains could be non-motile. Moreover, *C. difficile* isolates of RT 017 and 046 are weakly motile compared to the reference strain 630 [12].

## 2. Regulation of Gene Expression in Flagellar Locus

Flagellum biosynthesis is known to be linked to several cellular processes other than mobility, including toxin production, sporulation, adhesion, and metabolism [13]. Thus, it is not a surprise that numerous regulatory mechanisms of gene expression occur on the flagellar locus of *C. difficile*.

One of the most important is a mechanism of phase variation via DNA inversion, also called the “flagellar switch”. Described first by Anjuwon-Foster et al. in the R20291 strain, a member of the PCR ribotype 027, which included hypervirulent emergent strains, it was the fourth inversion site (Cdi4) identified in *C*. *difficile* among a number of eight inversion sites [14,15]. The flagellar switch is a 154 bp, invertible DNA sequence flanked by 21 bp inverted repeats that lies upstream of the early-stage flagellar gene operon (Figure 1). Firstly, bacteria with the flagellar switch in the orientation called the flg phase ON express the genes of the early operon encoding the flagellar hook, basal body, MS ring, motor, and other assembly proteins, along with the alternative sigma factor FliA, also known as SigD (or σ28 in Gram-negative bacteria). FliA activity is responsible for the gene transcription of the toxins via the direct control of an alternative sigma factor expressed by the *tcdR* gene, which is in the vicinity of the toxin genes [8,16]. Among the late operons is *flgM*, which is predicted to encode an anti-sigma factor that antagonizes FliA activity. Interestingly, FliA was shown to regulate the transcription of *flgM*, possibly as a retro-control of it [17]. Secondly, bacteria with the flagellar switch in the inverse orientation, called the flg phase OFF, were attenuated for flagellar and toxin gene expression and consequently are aflagellate and show decreased toxin secretion. The orientation of the flagellar switch may change during growth, and it was determined that a tyrosine recombinase, RecV, which also mediates an inversion at the *cwpV* switch (involved in the phase-variable expression of the largest member of cell wall proteins, CwpV), is responsible for the inversion of the flagellar switch [14]. Further, the difference in the gene expression between the two flagellar switch orientations has been explained by a post-transcriptional mechanism. Trzilova et al. established that the bacterial transcription terminator Rho factor is responsible for preferentially terminating the transcription of flg phase OFF mRNA within the 5′ leader sequence [18].

However, heterogeneity between strains exists concerning the possibility or frequency of the flagellar switch. For certain strains, the phase is locked in one orientation. For example, with clinical strains of ribotype 012, the flg phase OFF is predominant with variability in the frequency of the flagellar switch, whereas the contrary is found in the first sequenced strain named 630, also of ribotype 012 [19]. This has consequences for the toxin production and, consequently, the virulence of these strains.

Another regulation occurs with the presence of a c-di-GMP-binding riboswitch, Cd1, immediately upstream of the *flgB* open reading frame (Figure 1). Thus, it was demonstrated that high levels of c-di-GMP reduced the expression genes of the early operon as *flgB* and *FliA*, and then the flagellum biosynthesis and motility of *C. difficile* [20], and it was observed that the level of c-di-GMP is subject to negative regulation by a pleiotropic regulator, SinR, found in *C. difficile* [21]. Iron could also be an important element for the regulation of mobility. Berges et al. studied the iron regulation in *C. difficile* and showed that the ferric uptake regulator (Fur) induced or repressed the late and early operons, respectively. A *fur* mutant gave a reduced mobility to the bacteria, and a Fur box was found upstream on the first gene of the late operon [22]. Therefore, iron plays a role in flagellar regulation depending on its concentration found by the bacteria during its life cycle.

Another regulator of gene expression in the early operon was found by Edwards et al. RstA, which has the characteristics of members of the RRNPP (Rap/Rgg/NprR/PlcR/PrgX; formerly RNPP) family of proteins, acts by directly repressing the *flgB* promoter. These regulators are controlled by the direct binding of small quorum-sensing peptides [23,24].

In the late operon, *C. difficile* genomes harbor *crsA*, a gene that expresses the carbon storage regulator A (CsrA). This protein is a small homodimeric RNA-binding protein that regulates gene expression, mostly associated with virulence-related processes, by directly modulating the translation and stability of its mRNA targets. By overexpressing the protein in the 630 strain, Gu et al. showed that CsrA negatively regulates flagellin production and mobility and positively regulates toxin production and cellular adhesion [25]. Recently, Zhu et al. showed that CrsA negatively modulates the *fliC* expression post-transcriptionally in the R20291 strain, and that FliW, the flagellar assembly factor encoded by *fliW* positioned near *crsA*, is needed for FliC production but not for *fliC* gene transcription [26]. They suggested that FliW indirectly affects the *fliC* expression through the inhibition of CsrA post-transcriptional regulation (Figure 1). This partner-switching mechanism is found in *B. subtilis* and named “FliW-CsrA-Hag” (Hag is equivalent to FliC).

Finally, transcriptional analysis of *C. difficile* has demonstrated a differential expression of flagellar genes under heat shock conditions: *fliC* was down-regulated, the transcriptional level of *fliD* remained unchanged, and genes encoding the hook, rod, and basal body were up-regulated [27].

## 3. Structure of the Flagellum and Its Glycosylation

*C. difficile* strains carry mostly peritrichous flagella that differ in their numbers ([28] and Figure 2). However, the R20291 strain was found to be monotrichously flagellated with only a single flagellum present on its surface [29].

According to the genes found in the flagellar locus, the structure of the *C. difficile* flagellum is like the flagella of other Gram-positive species, with the classical three parts: the basal body, the hook, and the helical filament. Its mobility is powered by an ion motive force across the membrane that depends on the presence of the MotAB-type stator found in this bacterium. A proton gradient is therefore highly suspected [30]. However, particular glycosylation occurs on the *C. difficile* filament.

Flagellar filaments are composed of thousands of flagellin monomers. After extracellular export, the flagellin monomer subunits, which comprise four structural domains (D0–D3), are stacked into a helical filament with the conserved D0 and D1 domains facing inward into the filament core channel, while the variable D2 and D3 domains protrude outward from the core [31]. The sites of the glycosylation of flagellin monomers from a diverse number of bacterial species have been found within the two surface-exposed domains, D2 and D3 [32]. The precise biological role of this glycosylation process has yet to be determined. It may be required for the stability of the subunit–subunit interactions within the flagellar filament or, alternatively, for the efficient secretion of the flagellin monomer through the basal body apparatus [33]. Such modifications may be important in colonization, adhesion, autoagglutination, and finally, the subversion of the host immune defenses [34].

In *C. difficile*, two genetic organizations of the flagellar F2 glycosylation operon exist. They are named the type A and type B systems and, when present, these glycan biosynthesis genes are located near the flagellar structural gene *fliC*. In the first strain sequenced, the strain 630 with the type A organization, four genes are present downstream of a putative glycosyltransferase gene (CD0240) and encode a putative kinase (CD0241), a sugar nucleotide (nucleoside triphosphate) transferase (CD0242), a hypothetical open reading frame (ORF), and a second glycosyltransferase (CD0243). Twine et al. determined that *C. difficile* 630 produces flagellin glycosylated in O linkages at up to five sites, threonine or serine residues, with a HexNAc residue. They also demonstrated through mutagenesis that the glycosyltransferase gene CD240 is required for the proper assembly and consequent motility [33]. Faulds-Pain et al. studied more precisely the composition of the *C. difficile* 630 flagellin glycan and showed that its composition is formed of a single N-acetyl-β-glucosamine (GlcNAc) linked to a phosphorylated N-methyl-L-threonine at the oxygen at C3 of the sugar. They also examined the role of each gene of the locus A in post-translational modifications of the flagellin. They found that the open reading frame CD0241 encodes a kinase involved in the transfer of the phosphate to the threonine, the CD0242 protein catalyzes the addition of the phosphothreonine to the N-acetylglucosamine moiety, and CD0243 transfers the methyl group to the threonine. Mutations in these respective genes affected the motility and caused cells to aggregate to each other and become more adherent to abiotic surfaces. Also, by negative-stain transmission electron microscopy (TEM), it was observed that the flagellum filaments clump together [35]. Recently, by phosphoproteomic analysis, Hensbergen et al. found new modifications on the glycan with a phosphonate 2-AEP usually found in protozoan ciliates [36].

In the type B system of the F2 operon, upstream of the gene also found in the type A system encoding a predicted glycosyltransferase (GT1), there are six different genes encoding two glycosyltransferases (GT2 and GT3), two putative uncharacterized proteins, a putative carbamoylphosphate synthetase, and a putative ornithine cyclodeaminase [9]. Valiente et al. studied more precisely the functional role of the glycosyltransferases in this system. By mutagenesis, they showed that GT1 and GT2 are responsible for the sequential addition of a GlcNAc and two rhamnoses, respectively, and that GT3 is associated with the incorporation of a novel sulfonated peptidyl-amido sugar moiety. GT2 is also responsible for the methylation of the rhamnoses [37]. Whereas type B modification is not required for flagellar assembly, some mutations that result in the truncation or abolition of the glycan reduce the bacterial motility and promote autoaggregation and biofilm formation. Further, Bouché et al. described the structural characterization of novel flagellar glycans from PCR ribotype 027 strains of *C. difficile*, notably, the R20291 strain. They found a diverse, novel O-linked sulfonated peptidylamido-glycan moiety decorating the flagellin protein linked at the serine and threonine amino acids. It is important to note that a novel structural entity comprising a taurine-like nonreducing unit, which could be involved in evading the host immune system, was found at the carbohydrate moieties [38].

## 4. Role of Flagella during Pathogenesis

The disruption of the gut microbiota by an antibiotic treatment enables the colonization of the gut by *C. difficile* and provides a metabolic niche for the bacteria with a transient increase in the nutrient availability. The flagellar apparatus possibly contributes to its settling in the following ways: (a) by providing force-driven motility to nutrients, (b) by promoting adherence to host cells, (c) by promoting biofilm formation, and (d) by acting as an immunomodulator by triggering proinflammatory cytokines through the Toll-like receptor 5 (TLR5) signaling pathway.

### 4.1. Motility to Nutrients

The *C. difficile* genome contains a single predicted methyl-accepting chemotaxis protein within a putative chemotaxis operon, suggesting that *C. difficile* may regulate its flagellar motility in response to nutrients. Notably, Courson et al. demonstrated that the epidemic *C. difficile* strain R20291 exhibits regulated motility in the presence of components of intestinal mucus or glucose by modulating its swimming velocity and not its tumble frequency [39].

The motility of *C. difficile* in diverse media have been poorly studied. The mucosal layer in the lower intestine presents a certain viscosity that could impact the mobility of the bacteria and the contribution of the flagella. Indeed, with diverse *C. difficile* strains, Schwanbeck et al. characterized the swimming of the motile fraction in media with polyvinylpyrrolidone, a 360 MDa long-chained polymer, or with mucins, both of which can form gel-like structures. They observed that *C. difficile* displayed an unusual motility phenotype for peritrichous bacteria, with alternating, short, back-and-forth run phases of 0.5–3 bacterial lengths, corresponding to ~3–15 μm [40]. Battaglioli et al. studied the relevance of a rich environment in proline found during dysbiosis to *C. difficile* colonization. They showed the scavenging of proline by the bacteria, and that, with a strain mutated on an enzyme essential in the proline fermentation pathway, the colonization was drastically reduced [41]. Interestingly, a minimal medium defined by Schwanbeck et al. implied the presence of proline for the maximal and stable swimming motility of the bacterium. The amino acid cysteine, as well as a carbohydrate source, were indispensable [30].

### 4.2. Adherence to Host Cells

The role of the flagella in the adhesion of *C. difficile* to the epithelium is controversial. Tasteyre et al. showed that *C. difficile* crude flagella extracted as well as purified recombinant FliD and FliC proteins bound to immobilized axenic mouse cecal mucus but not to porcine stomach mucin in immunodot analyses [42]. Further, Dingle et al., using 630-strain mutants without flagella (mutated on the flagellin or on the flagellar cap protein), demonstrated that they adhered more efficiently to epithelial Caco-2 cells than the wild-type strain [43]. The complete lack of flagellin modification also significantly reduces the adhesion of *C. difficile* to Caco-2 intestinal epithelial cells [37]. But Baban et al. showed that flagellar mutants of strain R20291 adhered less than the parental strain in vitro [29]. They raised the question of the importance of the flagella in vivo and showed that a paralyzed mutant of strain R20191 outcompeted the aflagellated mutant in colonization and adherence, confirming the role of flagella in the adhesion process. However, flagella and motility were not needed for successful colonization with the strain 630Δerm, as there was no difference between the wild-type and flagellar mutants in diverse mouse models. It seems that the need for flagellation for adhesion and colonization depends on the strain and its regulation mechanisms of the flagellar locus.

Moreover, in another animal model largely used to study CDI, a hamster model, FliC or FliD were nonessential for cecal colonization, and flagellar mutant strains were more virulent, indicating either that flagella are unnecessary for virulence or that the repression of motility may be a pathogenic strategy employed by *C. difficile* in hamsters [43].

### 4.3. Role in Biofilm

The flagellum plays an important role in biofilm formation. For example, in *Pseudomonas aeruginosa*, flagella are not necessary for the initial attachment or biofilm formation, but the cell appendages play roles in the biofilm development and structure [44].

*C. difficile*, like other nosocomial pathogens, has also been found to produce biofilms in vitro and biofilm-like structures in vivo on the intestinal mucosal surfaces of infected mice [45]. Also, *C. difficile* biofilm is suspected to be involved in the persistence of the bacteria and responsible for the recurrence often found in *C. difficile* infections [46].

In *C. difficile* planktonic cells, low c-di-GMP resulted in high flagellum expression, while, during the biofilm formation, c-di-GMP in higher concentrations induced a decrease in the flagellum expression, as it was noticed for the flagellin gene *fliC*, which is significantly decreased in biofilm samples relative to planktonic samples [20,47]. However, Ðapa et al. observed a significant decrease in the biofilm accumulation for a *fliC* mutant compared to the wild-type strain after several days but not at earlier times. They suggested that flagella may be more important for later stages in biofilm formation [48]. In nature, biofilms are known to be usually formed by multiple microbial species. Engevik et al. reported that the addition of *Fusobacterium *nucleatum** enhanced *C. difficile* biofilm formation and extracellular polysaccharide production, and that the interaction of the two bacteria resulted in aggregation. Interestingly, the proteins responsible for the aggregation were found with RadD, an arginine-dependent adhesin for *F. nucleatum*, and with the flagellin FliC for *C. difficile* [49].

### 4.4. Acting as Immunomodulators by Triggering Proinflammatory Cytokines through the Toll-like Receptor 5 (TLR5) Signaling Pathway

Flagellin belongs to molecules containing a pathogen-associated molecular pattern (PAMP), which is recognized by the Toll-like receptor 5 (TLR5). The pathogenesis of *C. difficile* early in the course of infection is predominantly characterized by acute intestinal inflammation mediated by the inducible innate immune response. Indirectly, Jarchum et al. showed the potential role of flagellin in the development of the immune response. In fact, they demonstrated that the administration of purified Salmonella-derived flagellin prevents intestinal damage during *C. difficile* infection [50]. Yoshino et al. showed that FliC induced the activation of NF-κB in HEK293T cells, as well as p38 mitogen-activated protein kinase, and promoted the production of interleukin-8 and CCL20 of two other intestinal epithelial cell lines, HT29 cells and Caco-2 cells [51]. Also, Batah et al. showed that the interaction of flagellin and TLR5 predominantly activates NF-κB and, to a lesser degree, the MAPK pathways, leading to the up-regulation of the proinflammatory gene expression and the synthesis of proinflammatory mediators [52]. TLR5 is predominately present on the basolateral side of the epithelial cells, and it has been hypothesized that the toxins TcdB and/or TcdA of *C. difficile* allow the flagellin to reach TLR5 by disturbing the integrity of the cellular junctions. Such synergy of action was found by the authors of [6].

Flagellin could also be an immunomodulator via its recognition by a specific intracellular inflammasome. Chebly et al. assessed whether the flagellin of *C. difficile*, an extracellular bacterium, could internalize into epithelial cells and activate the NLRC4 inflammasome. This was demonstrated by confocal microscopy with the observation of the internalization of FliC tagged with a green fluorescent protein (GFP) into the intestinal Caco-2/TC7 cell line. Further, the activation of NLRC4, the cleavage of pro-caspase-1 into two subunits, p20 and p10, and also the cleavage of gasdermin D, were shown, suggesting that the caspase-1 and NLRC4 inflammasome activation by FliC contributes to the inflammatory process of *C. difficile* infection [53].

## 5. Immunogenicity of Flagella

To prevent *C. difficile* infection in contexts in which toxin-based vaccines are not clearly producing convincing results, the induction of specific antibodies directed against molecules involved in the adhesion/colonization processes by vaccination is an envisaged strategy. The flagellar components are promising candidates, and more particularly the components of the filament, the flagellin FliC, and the flagellar cap protein FliD, which are exposed extracellularly. FliC is well conserved in the N- and C-terminal regions, which are involved in the polymerization and secretion of FliC, whereas the central surface-exposed region is more variable (Figure 3). FliD is even better conserved among strains. Pechiné et al. showed that antibodies against both FliC or FliD are detected after CDI diagnosis and for the following 2 weeks. The antibody levels against FliC were significantly higher in hospitalized patients who did not develop CDI than in a CDI patient group, suggesting the possible protective role of such immune responses [54].

Ghose et al. showed that the recombinant protein FliC is immunogenic in mice, and that immunization with FliC is protective in a murine model of CDI following a challenge with a clinical hypervirulent *C. difficile* strain. Interestingly, mice that succumbed to the challenge had lower levels of anti-FliC IgG response compared to those that survived. Passive protection experiments using anti-FliC polyclonal serum in mice suggested that this protection is antibody-mediated. FliC was also able to afford partial protection in a hamster model of CDI [55]. Also, in a lethal hamster model, Bruxelles et al. showed that FliC encapsulated into pectin beads for colonic release and orally administered can confer significant protection, contrary to free FliC, while the intra-rectal and intra-peritoneal administration of FliC gave no protection [56].

A chimeric TcdA and TcdB toxin expressed in a nontoxigenic *C. difficile* strain was used to create an oral vaccine that can target both *C. difficile* toxins and colonization/adhesion factors. It induced systematic and mucosal antibody responses against the *C. difficile* flagellins FliC and FliD. The authors also found that an anti-chimeric toxin serum was significantly cross-reactive with FliC/FliD and two surface-layer proteins, the SlpA and Cwp2 proteins [57].

Finally, Razim et al. mapped the epitopes of both FliC and FliD in silico and by using the PEPSCAN procedure. They identified two promising epitopes on FliC, with one located in the TLR5-binding and activating region and one exposed on the polymerized flagellum structure. They also identified two promising well-exposed epitopes on FliD with potential protective properties according to VaxiJen analysis [58].

## 6. FliC as Vaccine Adjuvant

Several pathogen-associated molecular patterns (PAMPs) are tested as immune adjuvants and are typically administered with the target antigen for potentiating various immune responses. Bruxelles et al. assessed the ability of the *C. difficile* flagellin FliC to act as a mucosal adjuvant combined either with ovalbumin or the precursor of the surface protein S-layer protein SlpA of *C*. *difficile* as an antigen [59]. Two routes of immunization were tested in a mouse model: intra-rectal and intra-peritoneal. Results showed that FliC, as an adjuvant for immunization targeting ovalbumin, was able to stimulate a gut mucosal and systemic antibody response independently of the immunization route.

After a challenge in a mouse model, a significant decrease in *C*. *difficile* intestinal colonization was observed in groups immunized with FliC as an adjuvant co-administrated with the *C*. *difficile* S-layer precursor SlpA as an antigen compared to the control group.

## 7. Methods

Structured searches were carried out in the PubMed, Medline, ScienceDirect, Directory of Open Access Journals, SciELO, MedCarib, and Global Index Medicus databases without time and language restrictions. This is a systematic review following the guidelines of the PRISMA platform [60].

## 8. Conclusions

*C. difficile* is the most frequent cause of nosocomial diarrhea worldwide. The bacterium is an enteric pathogen for which the colonization of the gut following intestinal microbiota dysbiosis is essential. During this process, the flagellum production is tightly regulated, and many studies have focused on this regulation at the transcriptional or post-transcriptional levels showing various mechanisms. Further, the fact that flagellum expression and toxin gene expression are linked via the flagellar alternative sigma factor, SigD, indicates that the flagellum plays a central role in the pathogenicity of the bacterium. Therefore, precisely deciphering the course of all regulatory mechanisms of flagellum synthesis during the gastrointestinal tract colonization by this pathogen will help us to better understand the contribution of the flagella to the disease and envisage new therapeutic approaches.

Also, the chemotactic behavior of the bacterium and functional regulators for the control of mobility are still poorly studied. This may be an important field of future research to better evaluate which biochemical compounds or environmental conditions, such as temperature, oxygen levels, or osmolarity, play a role in the flagellum activity.

Furthermore, in the field of immunology, the issue of vaccination against *C. difficile* with the patient’s inability to generate a rapid, long-lasting, and protective response may be solved with a combination with extracellular components of the bacteria, such as the flagellin FliC or the flagellar cap FliD, which have been proven to improve the immunogenic properties of the bacterial antigens.

## Figures and Tables

**Figure 1 ijms-25-02202-f001:**
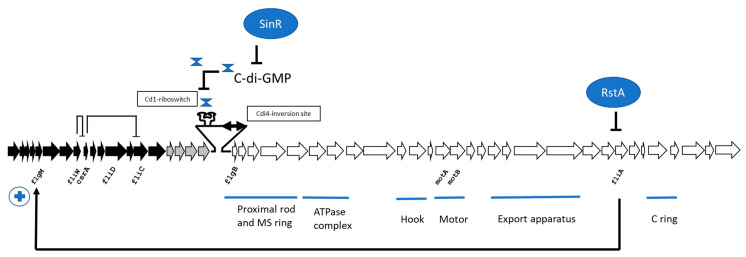
Genetic organization of the flagellar locus of *C. difficile*. The arrows show the extent and direction of the transcription of the genes with, in black, genes of the late operon (F3); in grey, genes of the glycan biosynthetic operon (F2-type 1 organization); and, in white, genes of the early operon (F1). Names of the genes cited in the text are indicated below their positions.

**Figure 2 ijms-25-02202-f002:**
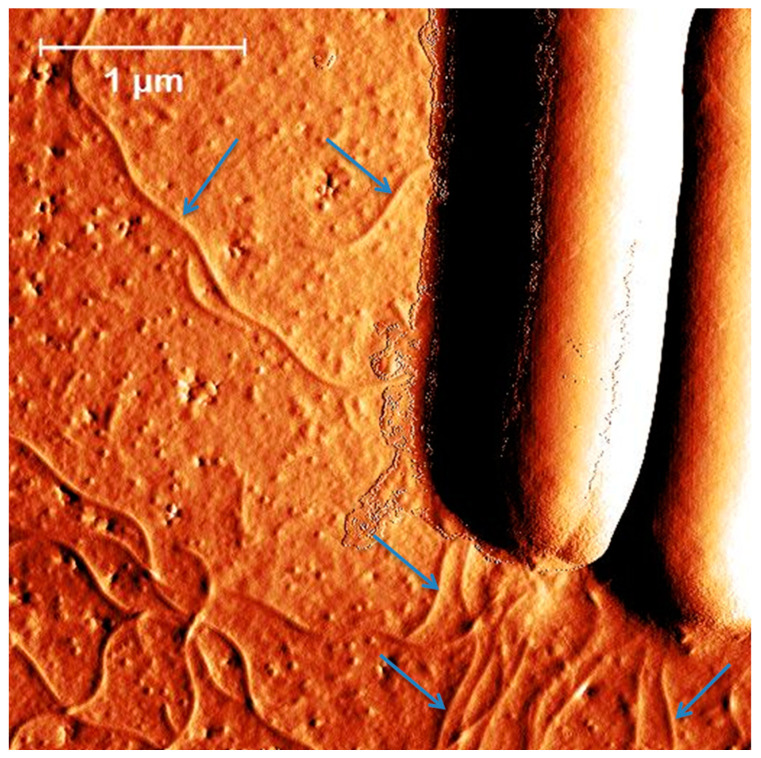
Atomic force microscopy (Innova, Brucker) of *C. difficile* 630 delta erm deposited on silicon wafer. Peritrichous flagella are indicated by blue arrows (personal image).

**Figure 3 ijms-25-02202-f003:**
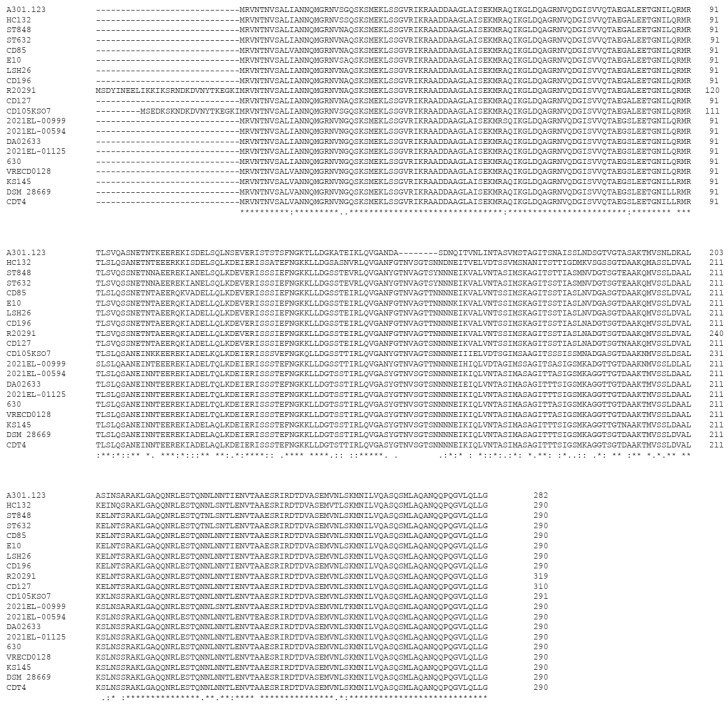
Alignments of FliC protein sequences of 20 *C. difficile* strains: A301.123 (GenBank: MCI9976946.1); HC132 (GenBank: NMS89690.1); ST848 (GenBank: UWI50438.1); ST632 (GenBank: UWD48927.1); CD85 (GenBank: EGT3785454.1); E10 (GenBank: CCK99127.1); lsh26 (GenBank: VFC60166.1); CD196 (GenBank: EQF88916.1); R20291 (GenBank: QPK95030.1); CD127 (GenBank: EQK93753.1); CD105KSO7 (GenBank: CZR95487.1); 2021EL-00999 (GenBank: MBY2229677.1); 2021EL-00594 (GenBank: MBY1102383.1); DA02633 (GenBank: HBG5846075.1); 2021EL-01125 (GenBank: MBY2081346.1); 630 (GenBank: NC_009089.1); VRECD0128 (GenBank: SJP36528.1); KS145 (GenBank: MDF3817077.1); DSM 28669 (GenBank: AXU63029.1); CDT4 (GenBank: AWH75966.1).

**Table 1 ijms-25-02202-t001:** Genes of the flagellar locus in the model laboratory strain *C. difficile* 630. ^a^ Locus_tags, gene names, and functions correspond to those indicated in the NCBI database Genbank (GenBank: AM180355.1).

**Locus_Tags ^a^**	**Gene Name ^a^**	**Function ^a^**	**Operon**
CD630_02270		Putative lytic transglycosylase	F1
CD630_02270		Conserved hypothetical protein	F1
CD630_02280	*fliN*	Flagellar motor switch protein	F1
CD630_02290	*flgM*	Negative regulator of flagellin synthesis (Anti-sigma-d factor)	F1
CD630_02300		Putative flagellar biosynthesis protein	F1
CD630_02310	*flgK*	Flagellar hook-associated protein	F1
CD630_02320	*flgL*	Flagellar hook-associated protein	F1
CD630_02330	*fliW*	Flagellar assembly factor	F1
CD630_02340	*csrA*	Carbon storage regulator homolog	F1
CD630_02350	*fliS1*	Flagellar protein	F1
CD630_02360	*fliS2*	Flagellar protein	F1
CD630_02370	*fliD*	Flagellar hook-associated protein	F1
CD630_02380		Conserved hypothetical protein	F1
CD630_02390	*fliC*	Flagellin C	F1
CD630_02400		Glycosyltransferase	F2
CD630_02410		Phosphoserine phosphatase	F2
CD630_02420		Conserved hypothetical protein	F2
CD630_02430		Conserved hypothetical protein	F2
CD630_02440		Glycerophosphotransferase	F2
CD630_02450	*flgB*	Flagellar basal body rod protein	F3
CD630_02460	*flgC*	Flagellar basal body rod protein	F3
CD630_02470	*fliE*	Flagellar hook–basal body complex protein	F3
CD630_02480	*fliF*	Flagellar M-ring protein	F3
CD630_02490	*fliG*	Flagellar motor switch protein	F3
CD630_02500	*fliH*	Flagellar assembly protein	F3
CD630_02510	*fliI*	ATP synthase subunit beta	F3
CD630_02520	*fliJ*	Flagellar protein	F3
CD630_02530	*fliK*	Flagellar hook-length control protein	F3
CD630_02540	*flgD*	Basal body rod modification protein	F3
CD630_02550	*flgE*	Flagellar hook protein	F3
CD630_02551	*FlbD*	Flagellar protein	F3
CD630_02560	*motA*	Flagellar motor rotation protein	F3
CD630_02570	*motB*	Flagellar motor rotation protein	F3
CD630_02580	*fliL*	Flagellar basal body-associated protein	F3
CD630_02590	*fliZ*	Flagellar protein	F3
CD630_02600	*fliP*	Flagellar biosynthesis protein	F3
CD630_02610	*fliQ*	Flagellar biosynthetic protein	F3
CD630_02620	*flhB*	Bifunctional flagellar biosynthesis protein	F3
CD630_02630	*FlhA*	Flagellar biosynthesis protein	F3
CD630_02640	*flhF*	Flagellar biosynthesis regulator	F3
CD630_02650	*flhG*	Flagellar number regulator	F3
CD630_02660	*fliA*	RNA polymerase sigma-28factor for flagellar operon	F3
CD630_02670		Putative flagellar protein	F3
CD630_02680	*flgG1*	Flagellar hook–basal body complex protein	F3
CD630_02690	*flgG*	Flagellar basal body rod protein	F3
CD630_02700	*fliM*	Flagellar motor switch protein	F3
CD630_02710	*fliN*	Flagellar motor switch phosphatase	F3
CD630_02720		Conserved hypothetical protein	F3

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
