# Peer review of "Clostridioides difficile Flagella"

_ijms, 2024, doi:10.3390/ijms25042202_

Round 1
Reviewer 1 Report
Comments and Suggestions for Authors
The manuscript titled "Clostridioides difficile Flagella" reviews current knowledge on the C. difficile flagellum. The manuscript covers a variety of topics such as genetic organization to therapeutic targets. Overall, the manuscript is easy to read. The figures are clear. There are text formatting issues such as font size and font style throughout the manuscript which can be easily be addressed. However, the manuscript needs more discussion on why this information is of value to the general audience. There is some discussion on immunogenicity of the flagellum and uses in vaccine development. But the manuscript does not provide a strong, detailed discussion on why its important to understand the flagellum and future research that can be done on this topic. The Introduction and Conclusion should go into more depth on the importance of the flagellum and flagellum research.
Reviewer 2 Report
Comments and Suggestions for Authors
The work presented to me for review is a review work on the occurrence of flagella in C. difficile. The work is well written and reliably presents a literature review on this topic.
However, I have a few comments:
1. In L-50, please enter the article from which the chart was borrowed.
2. In figure 2, please mark the scale and the flagellum with an arrow.
3. In L-290, please complete the sentence.
4. Remove figure 3, it is illegible and unnecessary in such work.
5. I suggest making a table and listing the set of flagellar and protein genes and presenting their characteristics.
6. Please make editorial corrections. The work is written in a different font, e.g., page 5 of the work.
Additional comments:
The authors of the work set themselves the goal of a genetic review of the flagellum and its various mechanisms of regulation, synthesis and glycosylation, its role in bacterial pathogenicity, as well as work on a vaccine. The goal set by the authors was fully and correctly analyzed in the work presented to me for review, based on a literature review.
Unfortunately, the work does not provide much of its own original information. The original is your own microscope photo (Figure 2). The FliC gene sequence is also included (Figure 3), the author does not indicate whether this is his original result.
The work is purely a review of the literature. The work is a compendium, a collection of information in one work on C. difficile flagella.
The authors selected the references appropriately, citing current works.
Figure 1 is correct and used in the appropriate place. Please include a citation of the graph in the text.
Figure 2 is an original, good quality photo. Please indicate and provide a detailed description of what the photo shows.
Figure 3 shows the fliC gene sequence. I believe that this photo does not contribute anything, I suggest removing it.
I suggest making a table and inserting it into the work, which would present all the genes, flagella proteins and their characteristics.
The conclusions are formulated correctly, but very general. Please provide detailed applications with responses related to the goal set at the beginning of the work.
To sum up, I believe that after the corrections the work is suitable for publication.
Round 2
Reviewer 1 Report
Comments and Suggestions for Authors
I thank the authors for responding to my comments and expanding on the introduction and conclusion. The authors did an excellent job providing details on importance of the C. difficile flagellum. I have no further comments to share.